# Solution Blown Nylon 6 Nanofibrous Membrane as Scaffold for Nanofiltration

**DOI:** 10.3390/polym11020364

**Published:** 2019-02-19

**Authors:** Ya Liu, Gaokai Zhang, Xupin Zhuang, Sisi Li, Lei Shi, Weimin Kang, Bowen Cheng, Xianlin Xu

**Affiliations:** 1State Key Laboratory of Separation Membranes and Membrane Processes, Tianjin Polytechnic University, Tianjin 300387, China; liuya8353@163.com (Y.L); 15620633941@163.com (G.Z); 2School of Textile Science and Engineering, Tianjin Polytechnic University, Tianjin 300387, China; m15009565769@163.com (S.L); shilei@tjpu.edu.cn (L.S); kangweimin@tjpu.edu.cn (W.K.); bowen15@tjpu.edu.cn (B.C);

**Keywords:** solution blowing, nylon 6 nanofibrous membrane, scaffold, nanofiltration

## Abstract

In this work, a nylon 6 nanofibrous membrane was prepared via solution blowing technology and followed hot-press as scaffold for nanofiltration. The structure and properties of the hot-pressed nylon 6 nanofibrous membrane (HNM) were studied the effect of hot-pressing parameters and areal densities. Then an ultra-thin polyamide (PA) active layer was prepared by interfacial polymerization on HNM. The effects of nanofibrous scaffolds on the surface properties of ultra-thin nanofiltration membranes and their filtration performance were studied. Results showed that the nylon 6 nanofibers prepared at a concentration of 15 wt % had a good morphology and diameter distribution and the nanofibers were stacked more tightly and significantly reduced in diameter after hot pressing at 180 °C under the pressure of 15 MPa for 10 s. When the porous scaffold was prepared, HNM with an areal density of 9.4 and 14.1 g/m^2^ has a better apparent structure, a smaller pore size, a higher porosity and a greater strength. At the same time, different areal densities of HNM have an important influence on the preparation and properties of nanofiltration membranes. With the increase of areal density, the uniformity of HNM increased while their surface roughness and pore size decreased, which is beneficial to the establishment of PA barrier layer. With areal density of 9.4 and 14.1 g/m^2^, the as-prepared nanofiltration membrane has a smoother surface and more outstanding filtration performance. The pure water flux is 13.1 L m^−2^ h^−1^ and the filtration efficiencies for NaCl and Na_2_SO_4_ are 81.3% and 85.1%, respectively.

## 1. Introduction

With the rapid development, the contradiction between the global shortage of drinking water and growing demand has also intensified. According to prediction of the United Nations, the global demand for water resources will increase by 30% by 2030, so it is particularly important to improve water treatment technology and achieve efficient water circulation [1]. Among the various technologies for water treatment, nanofiltration (NF) is considered to be a process with energy efficiency and environmental-friendliness, offering higher permeability than reverse osmosis (RO) and significantly better rejection than ultrafiltration (UF) [2,3,4]. In the past two decades, NF process has been gradually applied to water treatment for the removal of hardness, heavy metals and dissolved organic matters [5,6,7].

Generally, a nanofiltration membrane (NFM) is a composite of an ultra-thin active dense layer, a porous polymer scaffold and a membrane support. Among them, the ultra-thin active dense layer is a main functional layer formed by interfacial polymerization on a porous polymer scaffold [8,9]. At present, many studies have focused on introducing new monomers, surfactants and nanoparticles in the interfacial polymerization process to optimize the ultra-thin active dense layer with improved film properties [10,11,12]. For example, Babak et al. have developed a novel thin-film nanocomposite (TFN) nanofiltration membrane via interfacial incorporation of aminosilanized TiO2 nanoparticles [13]. Lewis et al. have prepared a new type of thin-film nanofibrous composite membrane (TFNC) by interfacial polymerization (IP) of piperazine (PIP) using ionic liquids (IL). The improved strategies for the above ultra-thin active dense layers have been well developed, significantly improving the film’s retention properties and environmental adaptability [14].

To form ultra-thin active dense layers, porous polymeric substrates are usually needed, which not only provides mechanical properties support for NFM but also has an important impact on the formation of ultra-thin active dense layers [15,16]. Recently, researchers have studied the effects of the same kind polymers with different physical properties, different polymers, blended polymers and modified polymers on ultra-thin active dense layers based on the types and properties of polymer raw materials [17,18,19]. Studies have shown that the surface hydrophilicity, surface roughness, pore size and porosity of the porous polymer scaffold are closely related to the thickness, roughness and cross-linking degree of the active dense layer formed by interfacial polymerization, which have an important influence on the nanofiltration performance of NFM [20,21,22]. At present, the porous polymeric substrates are still limited to the asymmetric porous membrane prepared by the non-solvent phase separation (NIPS) method, easily leading to many dead-end pores through which pure water cannot pass.

Nanofibrous membranes, which are composed of electrospun nanofibers with random orientation, have several unique characteristics such as a large surface area to unit volume ratio, high porosity (up to >80%), nanosize pores and fully interconnected pores [23,24,25]. With these properties, nanofibrous membranes provide abundant water pathways with low hydraulic resistance and have drawn remarkable attention in microfiltration with high water flux and low-energy consumption [26,27]. Recently, polyacrylonitrile (PAN) nanofibrous membrane was applied as nanofibrous scaffold for a high flux thin-film nanofibrous composite (TFNC) nanofiltration membrane and it exhibited over 2.4 times more permeate flux than conventional thin film composite (TFC) membranes with the same chemical compositions and the same retention rate [28]. The research indicated that the electrospun nanofibrous materials could act as an alternative functional supporting material instead of the polymeric phase-inverted support layer in liquid filtration [15,29,30,31]. However, the productivity of electrospinning is still a challenge in this stage which limits the industrial applications of nanofibrous membranes.

Solution blowing process is a novel nanofiber fabrication method with high-velocity gas flow as fiber formation driving forces, in which the spinning solution streams are blown to jets due to the shearing force at the solution/gas interface by high-velocity gas flow [32,33]. In our previous work, a polyvinylidene fluoride nanofibrous membrane was successfully solution blown using a multiorifices die and showed good performance in microfiltration with high pure water flux and high retention ratio against microparticles [34].

In this work, a solution blown nylon 6 nanofibrous membrane was hot-pressed as scaffold (HNM) for nanofiltration membrane, instead of NIPS polymer porous membrane. The conditions of spinning and hot-pressing were studied and the effects of structural properties of nanofibrous membrane under different areal densities on HNM and nanofiber-based nanofiltration membranes (NFNFM) were studied.

## 2. Materials and Methods 

### 2.1. Chemicals

Nylon 6 pellets were purchased from the Ube Industries, Ltd. (Tokyo, Japan). Anhydrous formic acid with analytical grade was purchased from Tianjin Miou Chemical Reagent Co., Ltd. (Tianjin, China). n-hexane with analytical grade was bought from Tianjin Sailboat Chemical Reagent Technology Co., Ltd. (Tianjin, China). m-phenylenediamine (purity 99.5%) was provided by Shanghai Aladdin Biochemical Technology Co., Ltd. (Tianjin, China). Triformyl chloride (purity 99.8%) was purchased from Shanghai Aladdin Biochemical Technology Co., Ltd. (Tianjin, China). PET spunbond non-woven fabrics with 40 g/m^2^ was supplied from Guangdong Yiyi Nonwovens Co., Ltd. (Tianjin, China).

### 2.2. Preparation of Nylon 6 Nanofibers by Solution Blowing

The dried nylon 6 pellets were dissolved in anhydrous formic acid at 105 °C for 8 hours to prepare nylon 6 spinning solutions with 12 wt %, 15 wt % and 18 wt %, respectively. A nanofibrous membrane was obtained using a solution blowing apparatus as shown in Figure 1. The equipment mainly includes high pressure gas source, suction system, porous spinning die, metering pump device, receiving system and spinning box. Each of the single holes in the multi-hole spinneret has two concentric channels as concentric fluid flow channels and the nylon 6 spinning solution is extruded through the internal passage while the high-pressure gas-flow is ejected through the external passage. When the spinning solution of the internal passage is extruded at a constant rate, the droplet is stretched to form solution jets under the shearing and drawing action caused by the high-pressure gas-flow at the spinneret and the nylon 6 nanofibers are formed after the solvent evaporated. The spinning parameters are shown in Table 1.

### 2.3. Preparation of HNM by Hot Pressing

The solution-blown nanofibers are highly curly and fluffy, so they are hot-pressed with a PET nonwoven to thin and dense porous layer. The hot pressing conditions used are at 180 °C for 10 s under the pressure of 0, 5, 10, 15 MPa, respectively.

### 2.4. Preparation of PA Active Layer by Interfacial Polymerization

The ultra-thin PA barrier layer is prepared and composited by interfacial polymerization on the HNM. As described in the literature [35], the steps are as follows. First, 2.0% of m-phenylenediamine (MPD) and 0.2% (*w*/*v*) of trimesoyl chloride (TMC) solution were prepared. Then, HNM was held using a clamp and immersed in MPD solution for 1 min. After finishing impregnation, the HNM was vertically dried in air for 5 min, followed by dipping in TMC solution for 30 s. Finally, the HNM was cured at 100 °C for 10 min and rinsed using DI water for future use.

### 2.5. Characterization

#### 2.5.1. Scanning Electron Microscopy (SEM)

The morphology of the nylon 6 nanofiber, HNM and NFNFM were observed using an SEM (Hitachi S-4800, Hitachi Limited Co., Ltd., Tokyo, Japan). Fiber diameter was determined from SEM micrographs using Image-Pro Plus (Ipwin32, Soft Imaging System, Pro 6.0, Media Cybernetics Co., Ltd., Washington, USA).

#### 2.5.2. Atomic Force Microscopy (AFM)

The surface morphology of HNM and NFNFM were determined by using an AFM (Agilent-S5500, Agilent Limited Co., Ltd., Palo Alto, CA, USA).

#### 2.5.3. X-ray Diffractometer (XRD)

The crystallinity of HNM before and after hot pressing was investigated using an XRD (D8discover, Bruker Limited Co., Ltd., Leipzig, Germany).

#### 2.5.4. Determination of Porosity (%) and Pore Size

The pore sizes of the HNM were measured with a 3H-2000PB Bubble Pressure Aperture Analyzer, which was able to probe pore sizes in the range of 0.02–500 μm. Pore size was measured for five times to calculate the average.

The porosity (ε) is the proportion of the volume of void to total volume of the HNM and calculated using the following Equation (1).
ε = (*M*_w_-*M*_d_)/(*A* × l × ρ_w_)(1)
In the formula, *M*_w_ is the weight of the wet membrane (g), *M*_d_ is the weight of the dry membrane (g), *A* is the effective area (cm^2^), *l* is the thickness (cm) of the membrane and ρ_w_ is the water density (g/cm^3^).

#### 2.5.5. Measure of Contact Angle

The hydrophilicity measurement was carried out by a contact angle goniometer (JYSP-180, Jinshengxin Testing Machine Co. Ltd., Beijing, China). The contact angle was measured for five times to calculate the average.

#### 2.5.6. Mechanical Characterization

Mechanical performance was determined with an Instron universal testing machine (Instron3369, Instron Limited Co., Ltd., Canton, MA, USA). The HNM had a size of 20 mm × 4 mm and the tests were performed at a strain rate of 1 mm/min. All the measurements were tested three times at room temperature.

#### 2.5.7. Characterization of NFNF

The flux and retention measurements of NFNFM were conducted under an assembled dead-end stirred cell system. The effective area of the membrane was 19.625 cm^3^ and during the measurement, a variable-speed diaphragm pump (DP-130, Xinxishan Industrial Co., Ltd., Wenzhou, China) was used to control the transmembrane pressure. The pressure was adjusted to 0.4 MPa to begin the flux and retention test. The water flux was calculated using Equation (2).
*J*_w_ = *V*/(*A* × Δ*t*)(2)
where *J*_w_ is the water flux (L m^−2^ h^−1^); *V* is the volume of permeated water (L); *A* is the active area of the membrane; and Δ*t* is the penetration time.

The aqueous feed solutions were prepared using NaCl or Na_2_SO_4_ salts separately. The conductivity of permeates was measured using a digital conductivity meter (DDSJ-308A, Inesa Limited Co., Ltd., Shanghai, China). The retention (R) was calculated using Equation (3).
*R* = (1 − *C*_p_/*C*_f_) × 100%(3)
where *C*_f_ and *C*_p_ are the conductivity of the feed and permeate concentrations.

## 3. Results and Discussion

### 3.1. Effect of Solution Concentration on Nylon 6 Nanofiber Forming

The concentration of the polymer solution affects the viscosity greatly, which is an important spinning factor. The solution viscosity is an external manifestation of the molecular chain entanglement of the polymers and proper chain entanglement is beneficial for spinning [36]. Figure 2 shows the morphology of nylon 6 nanofibers prepared at different solution concentrations. As can be seen in Figure 2, when the solution concentration is increased from 12 wt % to 18 wt %, the spinnability of the solution is increased and the fiber diameter tends to increase with increasing concentration. When the concentration of the solution is 12 wt %, nanofibers are formed with the evaporation of solvent in the solution jet flows under the airflow field but the fibers with the average diameter of 205.2 nm are fine and prone to filament breakage, filamentation and entanglement. As the concentration of the solution increased to 15 wt %, the fiber morphology was good without broken filaments, filaments and so forth. In addition, the fiber curl was reduced with no obvious entanglement and the average fiber diameter increased to 299.143 nm. When the concentration of the solution increased to 18 wt%, the average fiber diameter further increased to 363.999 nm with a wide diameter distribution and fine fiber entanglement occurred. This is because the increase in concentration drives the increased viscosity, the increased degree of molecular chain entanglement in the solution and the enhanced intermolecular force, so that more force is required for the solution jet drafting in the gas flow field. As a result, the fiber diameter becomes larger, the diameter distribution becomes wider, the degree of curling is lowered and the appearance of fine fibers causes entanglement. Therefore, 15 wt % solution concentration was chosen as parameter for subsequent work.

### 3.2. Effect of Pressure in Hot Pressing on HNM

Pressure is one of the key parameters in hot pressing. Different pressures were used to treat the same areal density (i.e., 6.1 g/m^2^) nylon 6 nanofiber to optimize conditions. Figure 3 shows the topography of the composite HNM under different pressures. The sample is looked loose under no pressure. As the applied pressure increases to 5 MPa, the fiber membrane becomes dense and partially melted plaques appear. The plaque area gradually increases with increasing pressure and the plaque range is maximum when the pressure reaches 15 MPa. The main reason is that the fibers accumulate more tightly under pressure and increase with pressure, accompanied by fiber deformation or melting, which causes plaques to appear, while shrinking the pores.

Hot pressing treatment has also an effect on the pore and fiber morphology of the composite nanofiber membrane. Figure 4 and Table 2 show the pore size distribution of HNM composited under the different pressures for the same areal density of nylon 6 nanofiber. As can be seen, as the pressure increases, the membrane pore size becomes smaller and the distribution becomes narrower. When the pressure is increased to 5 MPa, the pore size and pore size distribution change greatly and the average pore diameter decreases from 1.7437 μm to 1.6236 μm, which is due to the close packing of the fibers under pressure. When the pressure is increased to 10 MPa, the average pore size is reduced to 1.5947 μm and then 1.5363 μm at 15 MPa. The reason for this change is that not only the accumulation of fibers changes under high pressure but also the fiber morphology changes. As the pressure increases, the fibers change from the original loose state to a close packed state, achieving a reduction in the pore size. However, the reduction of the fiber accumulation state is limited. Continuing to increase the pressure has a smaller effect on the pore size but it is easy to partially melt or deform the fiber to reduce the pore size. Figure 5 shows the XRD patterns of HNM composited under different pressures. It can be seen that the crystallization peak and crystallinity of nylon 6 nanofiber change little before and after hot pressing, indicating that the fiber membrane melt change under hot pressing is small, so 180 °C, 15 MPa and 10 s are set as parameters for subsequent work.

### 3.3. The Effect of Areal Density on HNM

Four areal densities of nylon 6 nanofiber were utilized to combine with PET non-woven fabric to form HNM in order to study the effect of areal density on HNM. Figure 6 shows the AFM plot and roughness for HNMs with different areal density nylon 6 nanofiber. It shows that different areal densities have an obvious influence on the apparent structure and roughness of HNM. With the increase of areal density, the surface roughness of the film decreases and the flatness of the apparent structure increases. When the nanofibers weighed 4.3 and 6.7 g/m^2^, the apparent structure fluctuates greatly and the average roughness is 187.602 nm and 170.486 nm with a large mean square error, respectively. When the areal density of nanofibers increased to 9.4 and 14.1 g/m^2^, the apparent structure was flat and the average roughness decreased to 146.071 and 138.787 nm, respectively and the mean square error also decreased. The reason for this difference is that the change in areal density has an effect on the uniformity of the web. As the areal density increases, the uniformity of the web increases, so that the apparent structure after hot pressing is flat and the roughness is lowered.

The effect of different areal densities on HNM is also reflected in pore size, porosity and hydrophilicity. Figure 7 shows the pore size distribution of different areal densities of HNM. The different areal densities of HNM show good regularity. With the increase of fiber areal density, the pore size decreases, the pore size distribution becomes narrower and the porosity becomes larger. As summarized in Table 3, when the nanofiber areal density is 4.3 g/m^2^, the average pore diameter of the prepared porous scaffold is 1.7102 μm. The average pore diameter of the porous scaffold decreases to 1.4632 μm (6.7 g/m^2^), 1.3580 μm (9.4 g/m^2^) and 1.0624 μm (14.1 g/m^2^) as the areal density increases and the difference between the maximum aperture and the minimum aperture is also gradually decreasing from 1.3087 μm to 0.5215 μm. At the same time, hot pressing composite has little effect on the hydrophilicity of HNM. As shown in Table 3, the contact angles slightly decreased from 55^°^ to 51^°^ with the increased areal density HNM from 4.3 g/m^2^ to 14.1 g/m^2^. Comparative porosity analysis found that this difference was due to the fact that the porous scaffold with areal density of 9.4 and 14.1 g/m^2^ contained more nanofibers and had higher porosity for water penetration.

Figure 8 shows the tensile fracture of different areal density HNM. Since the HNM is supported by the same PET nonwoven fabric, the strength of the four porous scaffold is not much different. The HNM strength with areal densities of 4.3 and 6.7 g/m^2^ was similar to that of pure PET non-woven fabric and the HNM strength with areal density of 9.4 and 14.1 g/m^2^ was slightly increased. This is because, during hot pressing, the nanofibers will be filled into the pores of the non-woven fabric under temperature and pressure conditions and the mechanical properties of the non-woven fabric may be deformed or melted to form a mechanical lock, so that the overall strength is improved.

### 3.4. Effect of HNM on NFNFM

The nylon 6 nanofiber composite PET non-woven fabric is used as the porous scaffold and the nanofiltration membrane is prepared by interfacial polymerization. In this process, the MPD solution adheres to the pores of the nanofibers during the impregnation of the aqueous phase solution. When the porous membrane is in contact with the organic phase solution, the MPD and TMC will polymerize at the interface of the two phases to form a polyamide barrier. The porous membrane as a place where the interface polymerization occurs has an important influence on the polymerization reaction. Therefore, four different areal densities of HNM were used to prepare NFNFM and study its effect on the PA barrier layer. Figure 9 displays the SEM images of the surface and cross-sectional view of a PA barrier layer prepared from different areal densities of HNM and the surface properties of the PA barrier layer were summarized in Table 4. The surface morphology appeared to be so called ridge- and valley structure, which is similar to the previously published results [37]. It clearly shows that the PA barrier layer gradually becomes smooth and thick as the areal density of HNM increases. When the areal density of the scaffold was 4.3 g/m^2^, the PA barrier layer was successfully prepared, whose surface fiber traces were obvious and enrichment points appeared. In the SEM image with high imagination (Figure 9(a2)), it was found that the PA layer was tightly attached to the fiber and the surface was rough and somewhat shaped with an average roughness of 37.5 nm. The cross-sectional view (Figure 9(a3)) reflects the thin PA layer with approximately 69.73 ± 5.4 nm thickness. When the areal density of the scaffold was increased to 6.7 g/m^2^, the PA film layer became thicker with thickness of about 72.52 ± 3.2 nm and the surface fiber traces were lightened with roughness of only 32.9 nm. When the areal density of the scaffold was increased to 9.4 g/m^2^, the surface roughness of the PA layer was further reduced to 24.2 nm and the fiber traces were lightened, showing a smooth surface (Figure 9(c2)). In its high imagination SEM image (Figure 9(c3)), the thickness of the PA layer is increased to 79.85 ± 2.1 nm. When the areal density is further increased to 14.1 g/m^2^, the surface properties of the PA layer are similar to that at 9.4 g/m^2^, with an average roughness of 23.7 nm but the fiber trace is lighter and the film thickness is 86.21 ± 2.9 nm. This is due to the fact that the HNM with larger areal densities is more uniform with smaller pores and higher porosity, which favors the uniform formation of the thin film and may adopt a more cross-linked and packed (chain stiffness) structure with decreased chain mobility [38]. In addition, the small pores with high porosity makes the MPD aqueous phase solution more abundant. Thus, during interfacial polymerization, more MPD monomers would erupt from the saturated pores of the substrate and react with the TMC to produce PA film that was denser and more cross-linked. As a result, forming thicker PA layer, which was very helpful to enhance the stability of polyamide layer on nylon-6 surface [21,39,40]. Meanwhile, as shown in Table 4, the change in the areal densities has no influence on the hydrophilicity of the formed PA layer.

In addition, the PA barrier layer formed by interfacial polymerization has an important influence on the nanofiltration performance of the membrane. As shown in Table 5, NFNFM prepared by HNM with different areal densities have different filtration performance. Firstly, the filtration performance of NFNFM prepared by 4.3 and 6.7 g/m^2^ HNM is similar in pure water flux, reaching 14.9 and 14.7 L m^−2^ h^−1^, respectively. However, the water flux of the nanofiltration membranes with thicker PA layer prepared by 9.4 and 14.1 g/m^2^ HNM is lower than the first two, reaching 13.2 and 13.1 L m^−2^ h^−1^, respectively. Similarly, the fluxes for NaCl and Na_2_SO_4_ solutions have similar conclusions and the NFNFM prepared by 4.3 and 6.7 g/m^2^ HNM has a higher solution flux. In contrast, the NFNFM prepared by 9.4 and 14.1 g/m^2^ HNM has a higher retention efficiency for NaCl and Na_2_SO_4_. Especially, the NFNFM prepared by the 14.1 g/m^2^ HNM has the highest retention efficiency for NaCl and Na_2_SO_4_, respectively, which is 81.3% and 85.1%. The main reason for these differences is that there is a significant difference in the PA layer formed on the porous layer. The PA/PTFE nanofiltration membrane prepared by phase-transformed method has a retention rate of 95.6% for Na_2_SO_4_ under a flux of 6.5 L m^−2^ h^−1^ at 0.5 MPa. Compared with the electrospun nanofiber scaffold, the nanofiltration membrane prepared by solution blowing nanofiber porous scaffold has improved flux and retention [35]. However, compared with the traditional porous polymer scaffold, the nanofiltration membrane prepared with the solution blown nanofibrous membrane as scaffold has a great advantage in flux but the retention rate is slightly lower, as previous reported [9,41].

## 4. Conclusions

In this work, nylon 6 nanofibrous membranes were prepared based on solution blowing process and their structural properties under different hot-pressing pressures and different areal densities were studied. Ultra-thin polyamide active layer was prepared on HNM by interfacial polymerization method and the effect of different porous supports on the apparent properties of nanofiber-based nanofiltration membrane (NFNFM) and the filtration performance of monovalent and divalent electrolyte aqueous solutions was investigated. It was found that the nylon 6 nanofibers prepared at a concentration of 15 wt % had a good apparent morphology and diameter distribution and the nanofibers were more compactly stacked with significantly reduced pore size after being pressed at 180 °C and 15 MPa for 10 s. After preparation of the porous scaffold, HNM with an areal density of 9.4 and 14.1 g/m^2^ has a better apparent structure, a smaller pore size, a higher porosity and a greater strength. At the same time, different areal densities of HNM have an important effect on the preparation and properties of NFNFM. With the increased areal densities of HNM, the uniformity increases while the surface roughness and pore size decrease, which is beneficial to the formation of PA barrier layer. Especially, the surfaces of the nanofiltration membranes prepared at 9.4 and 14.1 g/m^2^ of HNM were smoother and the filtration performance was more prominent than those membranes prepared with lower areal densities of HNM (e.g., 4.3 and 6.7 g/m^2^). The NFNFM prepared by the 14.1 g/m^2^ HNM has the pure water flux of 13.1 L m^−2^ h^−1^ and the retention efficiency of 81.3% and 85.1% for NaCl and Na_2_SO_4_, respectively. Primary results indicated that solution blown nylon 6 nanofibrous membranes are promising candidates as scaffold for new high-performance nanofiltration membranes due to their high flux and ion rejection. It is believed that fabrication of NFNFM membranes based on laminated HNM scaffolds is promised the suitability of the large-scale industrial production.

## Figures and Tables

**Figure 1 polymers-11-00364-f001:**
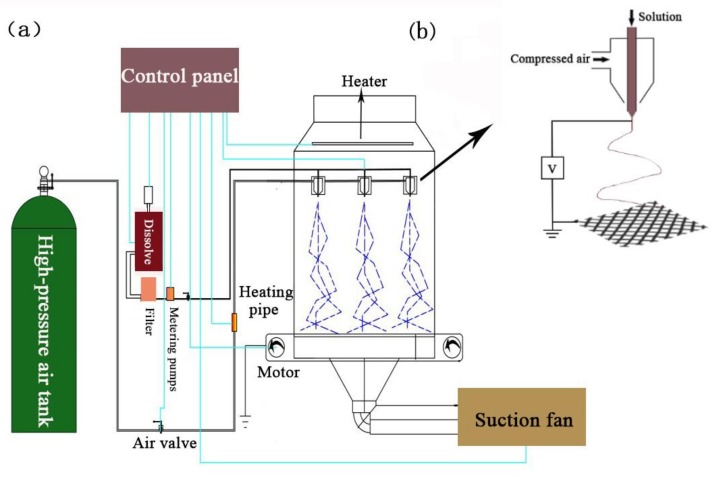
(**a**) schematic diagram of solution blowing device; (**b**) schematic diagram of single hole principle.

**Figure 2 polymers-11-00364-f002:**
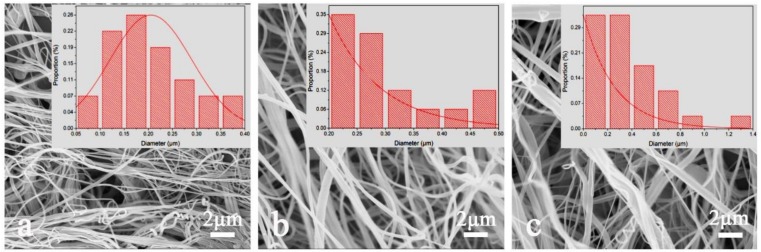
Scanning electron microscopy (SEM) images of nylon 6 nanofiber at different solution concentrations, (**a**) 12 wt %, (**b**) 15 wt % and (**c**) 18 wt %. Inset are their size distribution curves.

**Figure 3 polymers-11-00364-f003:**
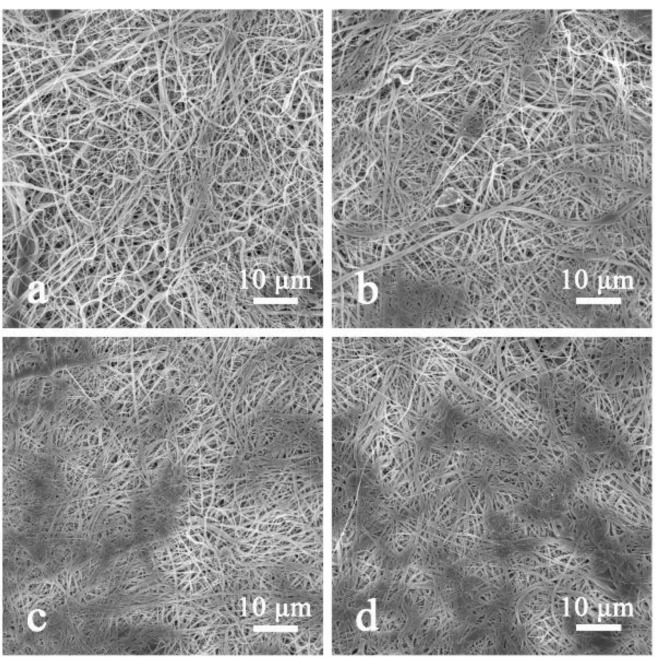
SEM images of composite hot-pressed nylon 6 nanofibrous membrane (HNM) under different pressures in hot pressing: (**a**) 0 MPa, (**b**) 5 MPa, (**c**) 10 MPa and (**d**) 15 MPa.

**Figure 4 polymers-11-00364-f004:**
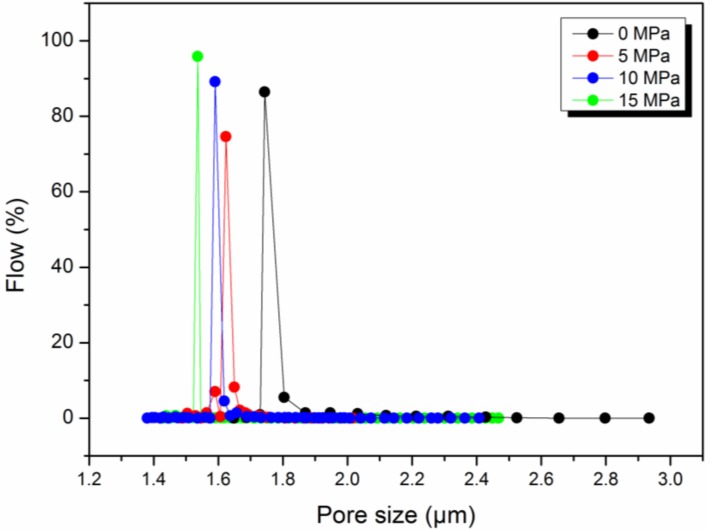
Pore size distribution of composite HNM under different pressures in hot pressing.

**Figure 5 polymers-11-00364-f005:**
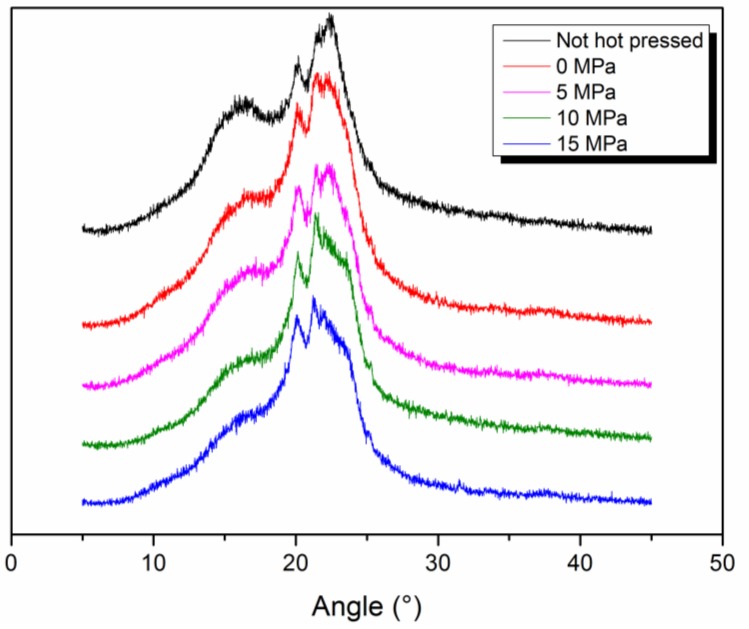
X-ray diffraction (XRD) curve of composite HBM under different pressures in hot pressing.

**Figure 6 polymers-11-00364-f006:**
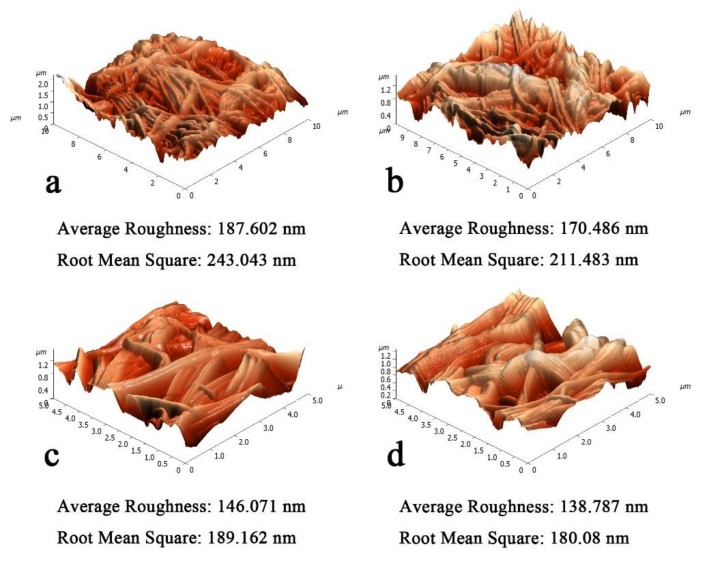
Apparent structure diagram (AFM) of different areal densities of HNM: (**a**) 4.3 g/m^2^, (**b**) 6.7 g/m^2^, (**c**) 9.4 g/m^2^, (**d**) 14.1 g/m^2^.

**Figure 7 polymers-11-00364-f007:**
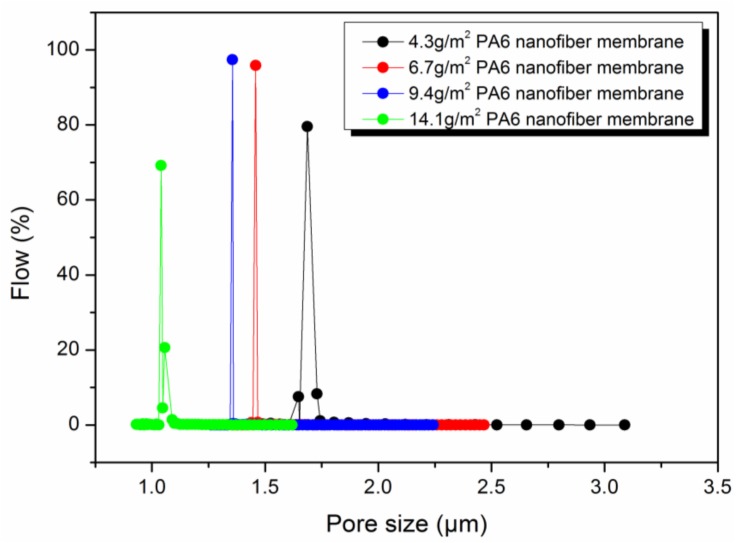
Pore size distribution of different areal densities of HNM.

**Figure 8 polymers-11-00364-f008:**
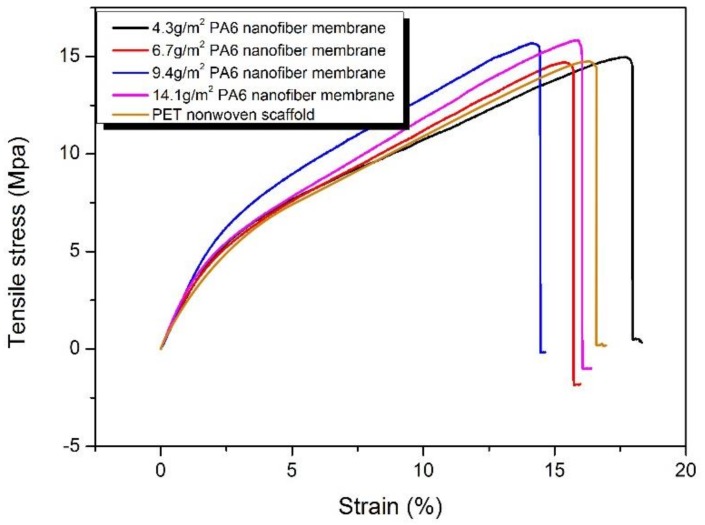
Tensile fracture curve of different areal densities of HNM.

**Figure 9 polymers-11-00364-f009:**
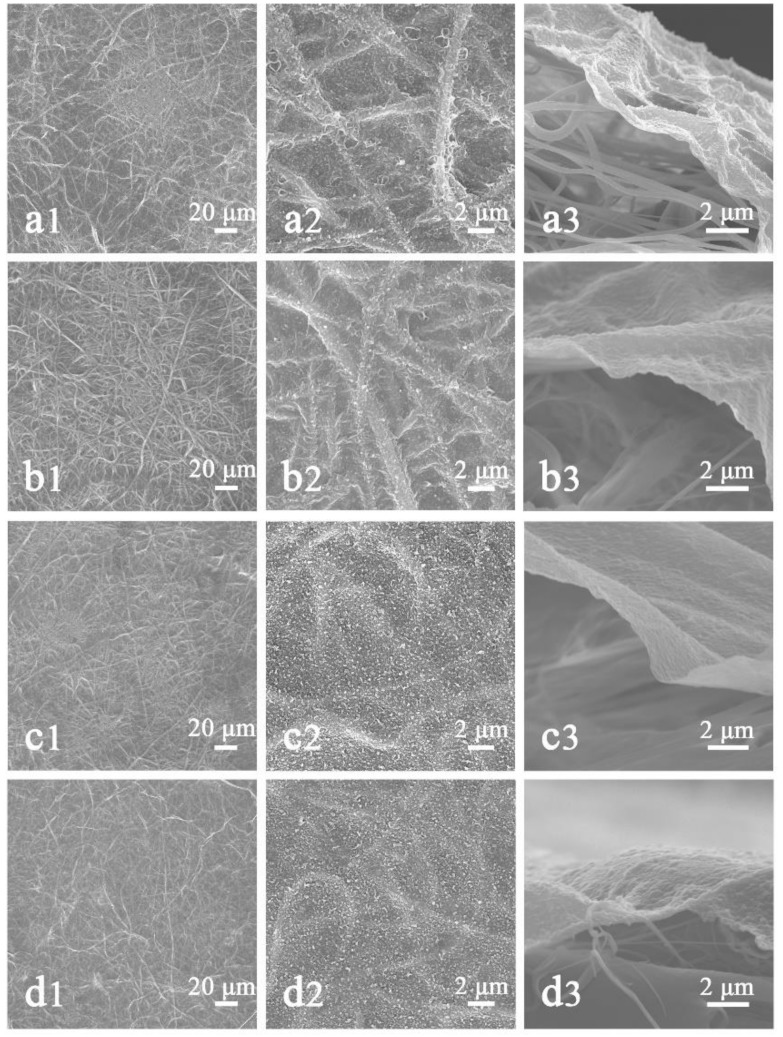
SEM images of surface and cross-section of NFNFM prepared by different areal densities of HNM: (**a1**)–(**a3**) 4.3 g/m^2^, (**b1**)–(**b3**) 6.7 g/m^2^, (**c1**)–(**c3**) 9.4 g/m^2^, (**d1**)–(**d3**) 14.1 g/m^2^

**Table 1 polymers-11-00364-t001:** Spinning parameters.

Process Parameters	Numerical Value
Metering pumps	17.5 r/min
Step size	0.6 ml/r
Cabinet temperature	45 °C
Inlet air temperature	60 °C
Metering pump temperature	30 °C
Dissolving kettle temperature	40 °C
Drafting wind pressure	2 bar
Auxiliary voltage	4 kV
Receiving distance	70 cm

**Table 2 polymers-11-00364-t002:** Pore size of composite HNM under different pressures in hot pressing.

Pressures	Maximum Pore Size (μm)	Mean Pore Size (μm)	Minimum Pore Size (μm)
0 MPa	2.5243	1.7437	1.6511
5 MPa	2.3154	1.6236	1.5043
10 MPa	2.2208	1.5947	1.3953
15 MPa	2.1147	1.5363	1.4392

**Table 3 polymers-11-00364-t003:** Pore size, porosity and contact angle of different areal density HNM.

Areal Density (g/m^2^)	Maximum Pore Size (μm)	Mean Pore Size (μm)	Minimum Pore Size (μm)	Porosity (%)	Contact Angle (°)
4.3	2.7978	1.7102	1.4891	78.2 ± 2.6	55 ± 0.5
6.7	2.1438	1.4632	1.4392	79.5 ± 1	54 ± 0.5
9.4	1.9906	1.3580	1.2794	83.4 ± 1.5	51 ± 0.5
14.1	1.4531	1.0624	0.9316	85.0 ± 3	52 ± 0.5

**Table 4 polymers-11-00364-t004:** Surface properties of nanofiber-based nanofiltration mebrane (NFNFM) prepared by different areal densities of HNM.

Areal Density (g/m^2^)	Contact Angle (°)	PA Layer Thick-ness (nm)	Average Roughness (nm)	RMS Roughness (nm)
4.3	54 ± 0.5	69.73 ± 5.4	37.5	42.0
6.7	53 ± 0.5	72.52 ± 3.2	32.9	39.6
9.4	54 ± 0.5	79.85 ± 2.1	24.2	30.0
14.1	54 ± 0.5	86.21 ± 2.9	23.7	28.1

**Table 5 polymers-11-00364-t005:** Filtration performance of NFNFM prepared by different areal densities of HNM.

Areal Density (g/m^2^)	Flux (L m^−2^ h^−1^)			Retention (%)	
	Water	NaCl	Na_2_SO_4_	NaCl	Na_2_SO_4_
4.3	14.9	9.3	9.7	73.2	79.6
6.7	14.7	8.7	9.4	76.5	81.3
9.4	13.2	8.2	8.5	80.4	84.7
14.1	13.1	8.1	8.2	81.3	85.1

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
