# Peer review of "Solution Blown Nylon 6 Nanofibrous Membrane as Scaffold for Nanofiltration"

_polymers, 2019, doi:10.3390/polym11020364_

Round 1
Reviewer 1 Report
In this paper, a nylon 6 nanofibrous membrane was prepared via solution blowing technology and followed hot-press as scaffold for nanofiltration. The structure and properties of the hot-pressed nylon 6 nanofibrous membrane (HNM) were studied the effect of hot-pressing parameters and areal densities. Then an ultra-thin polyamide (PA) active layer was prepared by interfacial polymerization on HNM. The effects of nanofibrous scaffolds on the surface properties of ultra-thin nanofiltration membranes and their filtration performance were studied. However, after reading the paper carefully, I do not think that this paper should be published in Polymers due to the following comments
The technology using nylon 6 nanofiber is not new. Some related papers have been reported in the past:
-Solution blowing nylon 6 nanofiber mats for air filtration
-Solution blowing of poly(dimethylsiloxane)/nylon 6 nanofiber mats for protective applications
More advances in the technology in this topic need to be clarified.
In Section 2.1, please provide the country information of the chemicals.
Line 96, 'g/m2'.
In Section 2.2, why did you choose the 12 wt%, 15 wt%, and 18 wt% solutions? Only three concentrations have been studies, why 16 wt% concentration could not be the best?
In ρw, is 'w' the subscript?
In Figure 2, it is hard to see the bar graphs and please correct.
Line 196, 'Pressure is the key parameters...' pressure is one of the key parameters???
In Section 3, do you have any references for verify your results in this work?
In Table 5, 'Na2SO4'.
All the results in this paper are mainly the characterizations of the membrane. Please provide the water purification performance using the membranes. Any optimization could be conducted for this case?
Please unify the format of the units of the parameters.
Author Response
Dear Reviewer,
Thank you for giving us the chance to revise our manuscript.
We are truly grateful to the reviewer’s critical comments and thoughtful suggestions. We have modified the manuscript accordingly, and the detailed corrections are listed below point by point. We hope the manuscript could fit the standard of Polymers.
Thank you again.
Best regards,
Xupin Zhuang
Tianjin Polytechnic University
February 10, 2019

Reviewer 2 Report
It is very interesting and useful to develop nylon-6 based membrane for nanofiltration. The authors carried out systematic studies with detailed discussion to study the fabrication of the membrane and to evaluate its performance. The filtration ability of electrolyte solution is in good level. The experimental data were concluded clearly as well. The reviewer recommends the acceptance of this manuscript after minor revision.
1. Some very related work should be cited: Langmuir 2009, 25, 7368-7374; Journal of Membrane Science 2012, 417, 228-236. These filtration membranes were developed based on similar concept.
2. How was the polyamide layer stabilized on nylon-6 surface should be discussed. The polyamide layer exhibited highly crosslinked structre, which was very helpful to enhanced the stsbility. See details in review paper Materials Horizons, 2015, 2 (6), 567-577.
3. A few sentences outlook can be added in the end of the conclusion to highlight the significance of this study in academy and industry.
Author Response
Dear Reviewer,
Thank you for giving us the chance to revise our manuscript.
We are truly grateful to the reviewer’s critical comments and thoughtful suggestions. We have modified the manuscript accordingly, and the detailed corrections are listed below point by point. We hope the manuscript could fit the standard of Polymers. (Please see the attachment.)
Thank you again.
Best regards,
Xupin Zhuang
Tianjin Polytechnic University
February 10, 2019

Round 2
Reviewer 1 Report
Please improve the technical and Englishe writing.